# Joining of Oxide Dispersion-Strengthened Steel Using Spark Plasma Sintering

**Foad Naimi [1], Jean-Claude Niepce [1], Mostapha Ariane [2], Cyril Cayron [3], José Calapez [3], Jean-Marie Gentzbittel [3] and Frédéric Bernard [2,*]**

[1] SINTERmat, 7 Avenue Maréchal Leclerc, 21500 Montbard, France; foad.naimi@sinter-mat.com (F.N.); jc.niepce@sinter-mat.com (J.-C.N.)

[2] ICB UMR 6303 CNRS/ UBFC, 9 Av. A. Savary BP47078, 21078 Dijon, France; moustapha.ariane@u-bourgogne.fr

[3] GRENOBLE-ALPES-CEA-LITEN, 38054 Grenoble CEDEX 9, France; cyril.cayron@epfl.ch (C.C.); jose.calapez@cea.fr (J.C.); jean-marie.gentzbittel@cea.fr (J.-M.G.)

\* Correspondence: fbernard@u-bourgogne.fr; Tel.: +33-3-80-39-61-25

**Abstract:** Difficulties with joining oxide dispersion-strengthened (ODS) steels using classical welding processes have led to the development of alternative joining techniques such as spark plasma sintering (SPS). SPS, which is classically employed for performing sintering, may also be used to join relatively large components due to the simultaneous application of electrical pulsed current and uniaxial charge. SPS technology was tested by joining two ODS steel disks. The preliminary tests showed that it is necessary to control surface roughness before joining. Furthermore, the use of ground and lapped surfaces seemed to improve the quality of the interface. Tensile tests on two ODS cylinders joined using SPS were performed at 750 °C without any additives. Failure occurred away from the interface with a total elongation close to 50% and an ultimate stress of 110 MPa.

**Keywords:** ODS steel; SPS joining; interface; tensile tests

## 1. Introduction

Generation IV nuclear power plants need to increase efficiency and produce less radioactive material. Compared to current thermal-neutron reactors, the new generation of fast-neutron reactors can reduce the total radiotoxicity of nuclear waste by using all or almost all of the waste as fuel. Due to their excellent creep strength, corrosion resistance, and radiation resistance, ODS (oxide dispersion-strengthened) ferritic steels are good structural candidates for cladding and fuel applications in fast-neutron reactors [1–3]. The strengthening of these materials is due to the homogeneous distribution of nanosized oxide particles, which inhibits dislocation movement and stabilizes the grain boundary microstructure [4–6]. However, the joining of ODS alloys proves to be challenging. The classical fusion welding techniques such as electric arc welding or high energy fusion such as laser welding or electron beam welding, results in the agglomeration as well as coarsening of dispersed nanoparticles and a loss of strength in the joined material [7]. Solid-state welding processes such as inertia friction welding, friction stir welding, and pressure resistance welding have already been explored [8–10]. Mechanical testing and microstructure observation of ODS steels joined using solid-state diffusion bonding have also obtained encouraging results [11,12]. The SPS (spark plasma sintering) technique was employed in this study to join bulk ODS ferritic steel to itself without disrupting both the fine uniform dispersion of nanosized oxide particles and the grain structure. Joining was achieved by supplying an electrical pulsed current to the samples in order to cause rapid heating via the Joule effect, while simultaneously applying uniaxial pressure. The two cylindrical

ODS steel samples were encapsulated in a graphite tool, formed by a die and two punches used to transmit the applied load. The electrical pulsed current flows through the setup formed by the graphite assembly and the bulk sample. This leads to direct heating via the Joule effect in a very short time [13,14]. Recently, SPS was used as an alternative consolidation technique of ODS steel powder [15,16]. A few studies have dealt with bulk metallic material joined using SPS [16] and only one of them has been performed on an extruded 20 wt.% chromium ODS PM2000 sample (PM2000 is a commercial product of the Plansee Company), which exhibits a fine microstructure [17]. The use of SPS technology as a means of joining two recrystallized large grain ODS PM2000 disks is the subject of the present article.

## 2. Materials and Methods

SPS joining was performed on an FCT HPD-25 spark plasma sintering machine (FCT Systeme, Rauenstein, Germany), located in the MATEIS laboratory at INSA Lyon using an on/off pulse sequence of 10:10 ms. Two joining configurations were tested: one with and one without a graphite die (Figure 1a,b), for which two thermal distributions were simulated (Figure 1c,d) using a thermo-electrical Abaqus model. In Figure 1a, the temperature was monitored using a radial pyrometer that determines the temperature at the centre of the graphite die wall 2 mm from the sample. SPS-joining was performed using a graphite foil of 0.2 mm thickness inserted between the various system components. Addition of the graphite foil helps to preserve the SPS tools as well as improve the electrical contact between each SPS component. In Figure 1b, the IR pyrometer is placed in contact with the joining area. This latter configuration was selected in order to eliminate any contamination of the material with graphite. Moreover, the temperature measurement via the radial pyrometer was performed at the contact zone between the two disks, which is un-joined at the beginning of the SPS cycle.

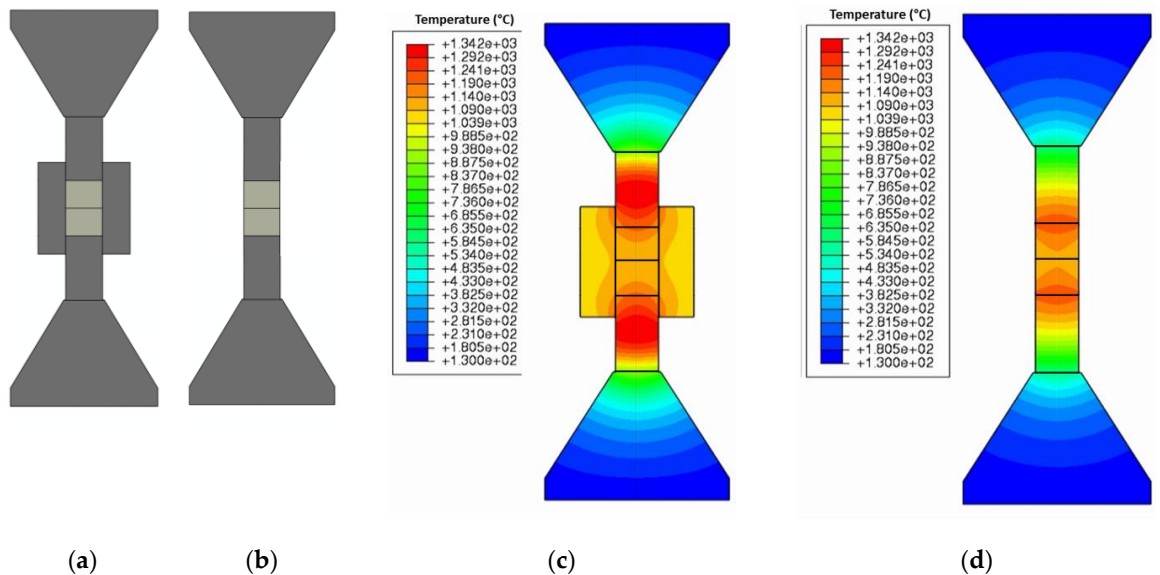

(**a**)　　　　　(**b**)　　　　　　　(**c**)　　　　　　　　　(**d**)

**Figure 1.** SPS configurations investigated in this study where (**a**) uses a die and (**b**) does not use a die. Numerical thermal mapping for both configurations where (**c**) represents the experimental setup with a die and (**d**) is without a die.

The thermal mappings for the two configurations previously described (Figure 1c,d) do not show a real difference regardless of the presence or absence of the SPS graphite die. Indeed, both thermal and electrical contact resistance between each component is assumed as perfect. Consequently, only the material properties can affect the flow of current. In our case, the sample and the die were both electrical conductors, therefore, a small proportion of current was diverted to the die, but the majority flowed through the sample (Figure 2).

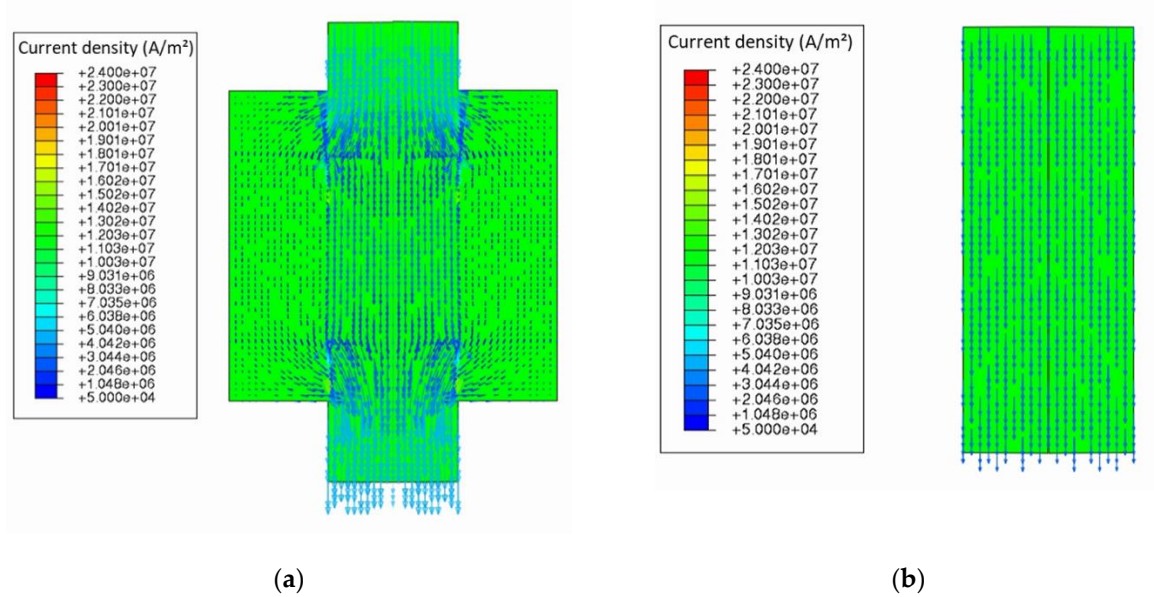

(**a**)                                          (**b**)

**Figure 2.** Current vector field for a sample (**a**) with a die and (**b**) without a die.

A recrystallized ODS PM2000 steel bar was delivered from Plansee AG, with a cylindrical diameter of 100 mm. PM2000 is a ferritic 19 wt.%, Cr–5.5 wt.%, Al–0.5 wt.%, Ti–0.5 wt.%, $Y_2O_3$ steel. The as-received microstructure was composed of large elongated grains (~few mm) parallel to the bar axis orientation, as shown in Figure 3.

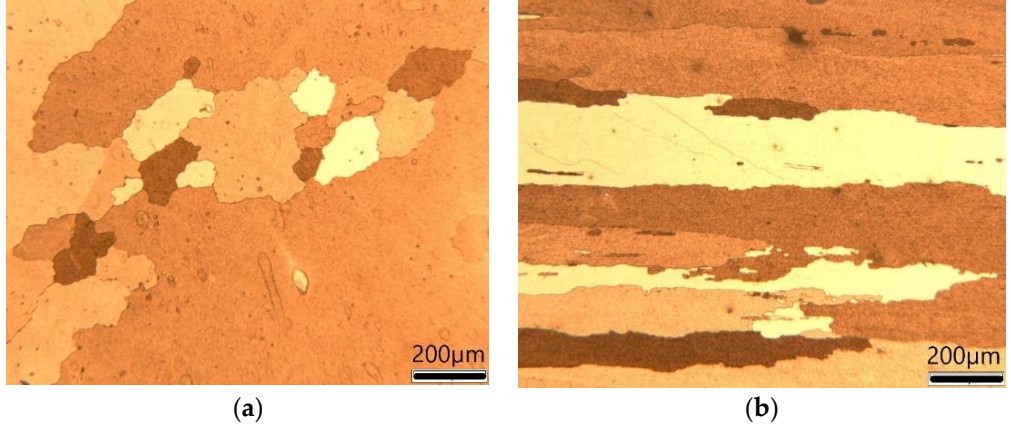

(**a**)                                          (**b**)

**Figure 3.** Optical micrographs of a PM2000 sample where (**a**) shows a transverse section and (**b**) depicts a longitudinal section.

Cylindrical samples of 10 mm diameter and 5 mm height were machined in the axis parallel to the bar axis. Only the orientation "transverse–transverse" of the join, shown in Figure 4, was tested.

In order to evaluate the SPS joining technology, a similar SPS procedure involving surface preparation, SPS environment, and thermal conditions was carried out without an additional interlayer (i.e., direct contact between parts to be assembled). SPS diffusion bonding of samples with varying surface roughness was investigated. Three levels of surface finish were tested corresponding to the following surface roughness values:

- Rough machined surface (Ra ~1.2 microns).
- Ground surface (Ra ~0.17 microns).
- Lapped surface (Ra ~0.04 microns).

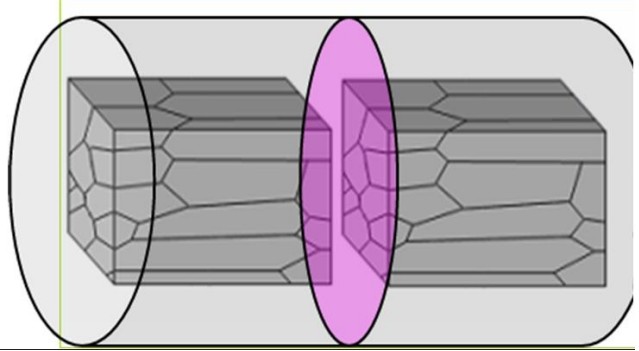

**Figure 4.** Diagram of the SPS joining orientation according to the sample orientation.

Several experimental SPS conditions, summarized in Table 1, were tested.

**Table 1.** Experimental conditions for SPS joining of ODS steel disks.

| 3 Sample Preparations | Heating Rate | Pressure | Temperature | Holding Time |
|---|---|---|---|---|
| Machined surface<br>Ground surface<br>Lapped surface | 150 °C/min | 20 MPa without a die to 64 MPa a with die | 950 °C to 1100 °C | 0 to 30 min |

In order to evaluate the possibility of performing a SPS-joining on ODS steel, two sample dimensions were investigated:

(i) On small disks (two disks with a 10 mm diameter and 5 mm thickness) to determine the best SPS-joining conditions. In this case, the classical SPS environment (graphite die and graphite foil called papyex®) was used. The main objective of these preliminary tests was to study the influence of the surface roughness on the quality of the interface. The quality of SPS welding, especially the quality of the SPS interface, was evaluated by studying the metallographic micrographs obtained after cutting the samples in the longitudinal orientation. Then, the samples were polished and chemically etched with a solution (20 mL HF + 50 mL HCl + 10 mL $HNO_3$ + 30 mL water).

(ii) On larger samples (two cylinders with a 20 mm diameter and 15 mm length) for the purpose of performing tensile tests at room temperature and at 750 °C in order to evaluate the mechanical resistance of the interface. Moreover, to achieve a clean interface without graphite contamination, SPS-joining was performed without a die using the lowest possible applied load of 16 MPa.

## 3. Results and Discussion

### 3.1. Small Disks: Influence of Surface Roughness

The optical micrograph shown in Figure 5, relative to the sample prepared with a machined surface and joined using SPS at 975 °C for 10 min, revealed large voids corresponding to the initial surface roughness (Ra = 1.2 μm) and large unbounded areas measuring up to 150 μm. Additional experiments showed that time and temperature did not remove the pores or reduce the size of defects at the joining interface.

Metallographic observations showed that SPS-joining of samples with a lapped surface were more encouraging. Only some areas remained un-joined or contained pores (Figure 6a). In addition, some cracks were observed at lower temperatures (950 °C, Figure 6a) and remained after joining at higher temperatures (1025 °C and 1050 °C, Figure 6b). However, some joints seemed perfect, as shown in Figure 6b. The observed interface was clean, without cavities, and with few small grains at the interface.

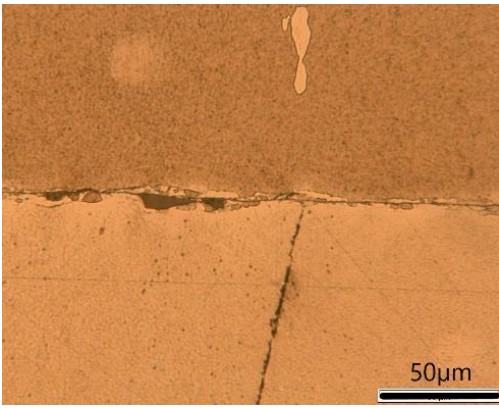

**Figure 5.** Micrograph of the bonding interface. SPS joined at 975 °C for 10 min from a sample with a rough machined surface.

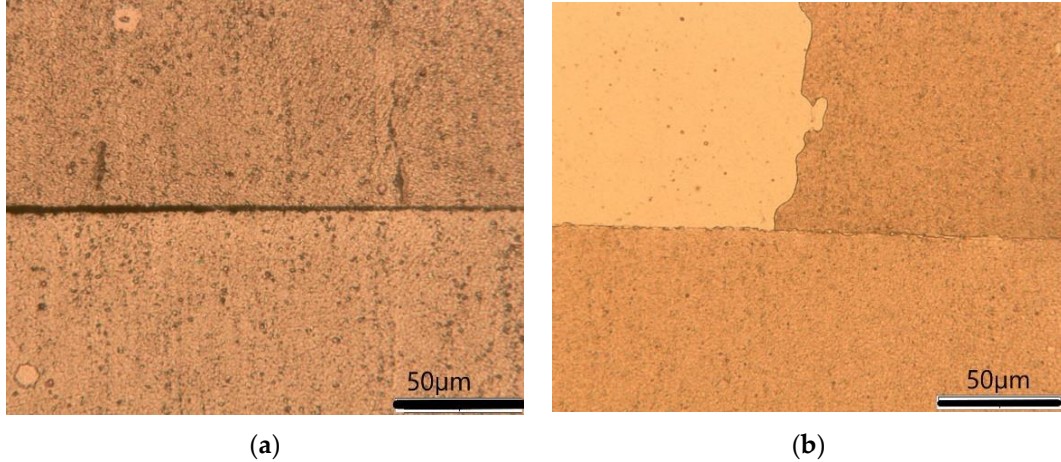

(**a**)             (**b**)

**Figure 6.** Metallographic observations of the interface in a joint made from two pieces with lapped surfaces, showing: (**a**) a poor-quality joint at lower temperature (950 °C) and (**b**) a good quality joint at higher temperatures (1025 °C and 1050 °C).

Metallographic results of the samples prepared with a ground surface were quite similar to those obtained using the SPS of a lapped surface. A satisfactory joint was obtained at 975 °C for 10 min although some large defects appeared. The initially large elongated grains did not cross the interface. Moreover, a strong crystal lattice distortion in the interfacial zone was observed in a 10 µm wide area on both sides of the interface using backscattered electron (BSE) imaging under a scanning electron microscope (SEM) (Figure 7). The presence of micron-sized grains at the interface could be observed.

Electron backscatter diffraction (EBSD) images (Figure 8a–c) revealed new micron-sized grains that were randomly oriented and are highlighted at the joining interface. These small grains were formed via recrystallization at the joining interface during SPS operation at 975 °C. Such small grains have also been observed on lapped surfaces, but their number and size were relatively small.

Consequently, the preliminary results confirmed the possibility of joining PM 2000 small disks using SPS. From previous tests, successful joining was only achieved with samples that had their surfaces lapped or ground. In contrast, SPS joining of samples with a machined surface did not lead to satisfactory joining. Too many defects (non-welded areas and cavities) remained, despite an increase in the SPS temperature or holding time. Distortion along the interface combined with the application of a high SPS temperature resulted in the recrystallization of new micrometric grains. In conclusion, these preliminary tests, despite the observed defects, have demonstrated the potential to join PM2000 samples using SPS. This work shows that the SPS temperature and the holding time have a slight

effect on the microstructure of the bonding interface. Bonding coarse grain to coarse grain is known to be difficult [18], however, the micrographic observations revealed some very clean joining areas. SEM and EBSD imaging of the well-bonded interface revealed recrystallized small micron-sized grains that were randomly oriented.

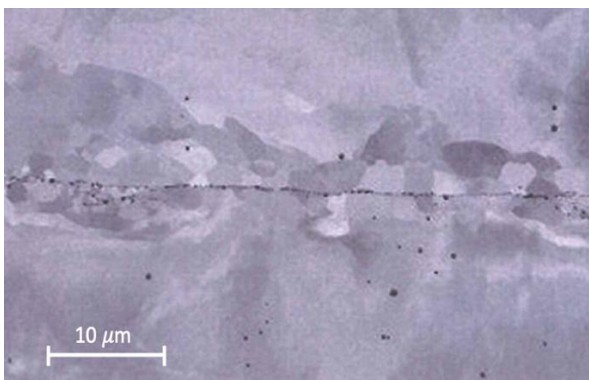

**Figure 7.** Scanning electron microscopy (SEM) image in backscattered electron (BSE) imaging of the interface between two joined disks with ground surfaces.

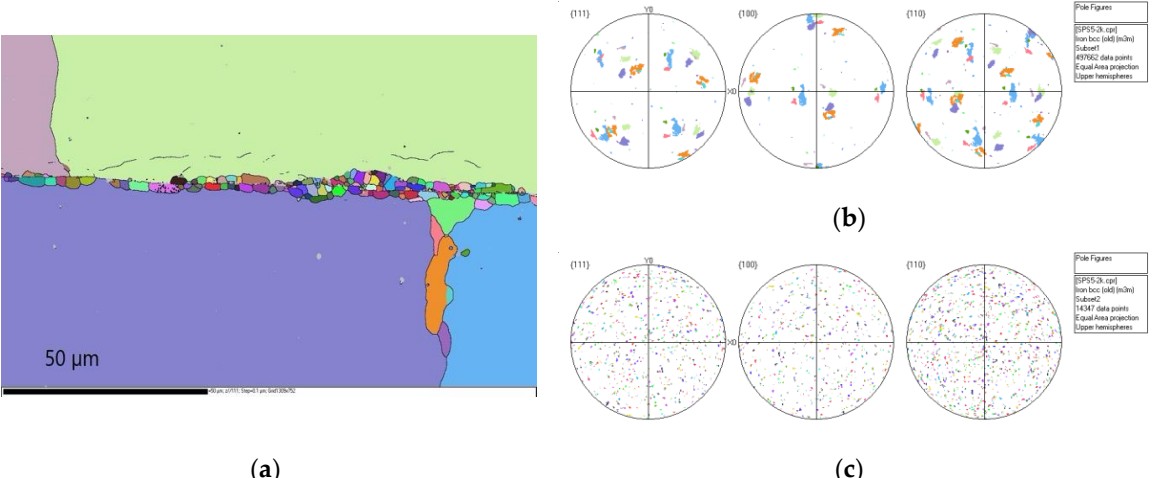

**Figure 8.** Interface between two joined disks where (**a**) shows the associated electron backscatter diffraction (EBSD) image, (**b**) reveals the pole figures of the large millimetric grains, and (**c**) represents the pole figures of the small micron-size grains at the interface. The same colour code is used in (**a**), (**b**), and (**c**).

### 3.2. Large Disks: Evaluation of the Mechanical Properties

Five large-size samples were joined with SPS using the welding conditions reported in Table 2. An example of a joined block obtained using SPS is shown in Figure 9.

**Table 2.** Summary of the SPS-joining conditions.

| Samples | Surface Preparation | SPS Conditions |
|---------|---------------------|----------------|
| 1 | Ground | 975 °C/20 min |
| 2 | Ground | 1050 °C/20 min |
| 3 | Ground | 1100 °C/10 min |
| 4 | Lapped | 1100 °C/3 min |
| 5 | Lapped | 1100 °C/10 min |

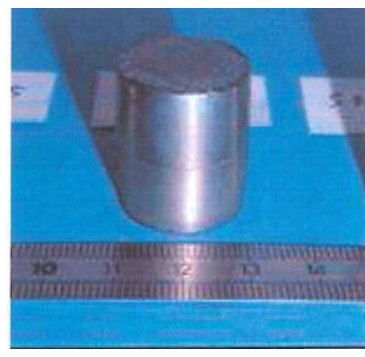

**Figure 9.** Example of a block joined using SPS.

The sampling of tensile specimens in each block was made using electrical discharge machining (EDM), according to the orientation defined in Figure 10. Two small-sized cylindrical tensile samples with a gauge length of 10.2 mm and a gauge diameter of 2.6 mm, along with two additional flat tensile specimens ($2 \times 2$ mm$^2$ section and 10 mm gauge length) were machined in each block. The welding bond failed during the machining of numerous tensile test specimens and thus a limited number of tensile specimens were available. Tensile tests were performed at room temperature and at 750 °C in air using an initial strain rate of $5 \times 10^{-4}$ s$^{-1}$, according to the NF-EN6892-1 and NF-EN6892-2 standards, respectively. At 750 °C, the tensile specimens were heated with a radiant furnace for 30 min prior to testing.

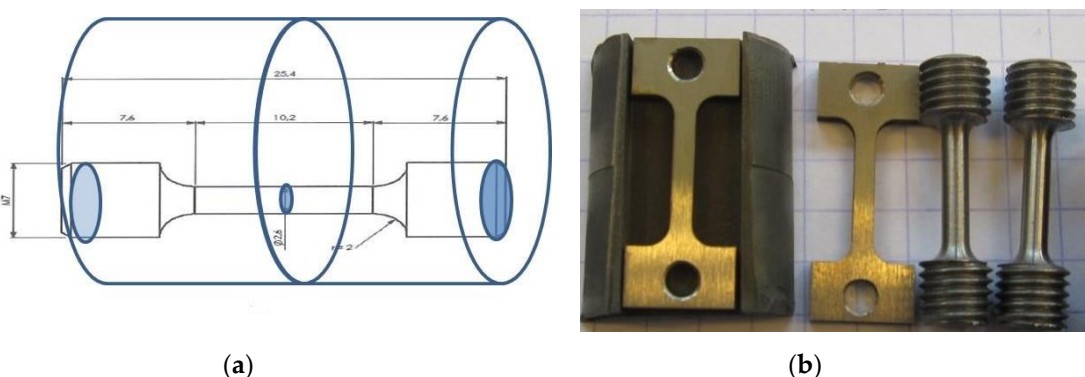

(**a**)  (**b**)

**Figure 10.** (**a**) Tensile testing samples were fabricated using electrical discharge machining (EDM) from the joined blocks. (**b**) Examples of flat and cylindrical tensile testing samples.

Only one tensile specimen from sample 1 could be tested at room temperature. The tensile specimen broke at the welded interface at a stress value of 282 MPa, a value much lower than the elastic yield strength of the parent material (Table 3). At room temperature, due to the anisotropic tensile behaviour of large textured grain, the strain mismatch at the bonding interface in such a small-sized tensile specimen induced high stress concentration at the welding interface.

At 750 °C, the PM2000 tensile behaviour was much more ductile and both tensile specimens extracted from Sample 1 failed at the interface and resulted in an ultimate tensile strength (UTS) of 110 MPa and 114 MPa associated with a total elongation of 6% and 8%, respectively. Sample 2 also demonstrated promising mechanical strength (~112 MPa), however, too many cracks were generated, leading to failure at the joined interface. The tensile rupture properties of all SPS-joined samples are listed in Table 3.

**Table 3.** Results of the tensile tests performed at 20°C and 750°C for the PM2000 SPS-joined disks

| Temperature | At 20 °C | | | At 750 °C | | |
|---|---|---|---|---|---|---|
| Samples | Ultimate Tensile Strength (MPa) | Total Elongation (%) | Failure | Ultimate tensile strength (MPa) | Total Elongation (%) | Failure |
| 1a | 282 | 0 | interface | 110 | 6.6 | Interface |
| 1b | | | | 114 | 8.4 | Interface |
| 2 | No tensile specimen available | | | 112 | 0.4 | Interface |
| 3 | No tensile specimen available | | | No tensile specimen available | | |
| 4a | 200 | 0 | | 101 | 53.0 | PM 2000 |
| 4b | 534 | 2.9 | | 116 | 44.0 | PM 2000 |
| 5 | No tensile specimen available | | | No tensile specimen available | | |

From these results, a higher temperature of 1100 °C and lapped surface preparation was chosen to perform SPS-joining of Sample 4 and Sample 5 (Table 2). Four tensile specimens could be machined from Sample 4, while the entire specimen from Sample 5 broke during final machining. Improved mechanical results compared to the previous were obtained at 20 °C and 750 °C (Table 3). The bonding strength at room temperature of one tensile specimen reached a value of 534 Ma, associated with an elongation of 2.9% of the base material and rupture was located at the welding interface. The second specimen broke at a lower value (200 MPa) and without plastic deformation. After a tensile test at 750 °C, specimens achieved an ultimate tensile strength of 101 and 116 MPa and a total elongation of 53% and 44%, respectively, and they fractured in the bulk material rather than at the joint (Figure 11a,b). In addition, Figure 11b presents the stress–strain curves for the specimens taken from Sample 4 and the as-received PM2000 alloy.

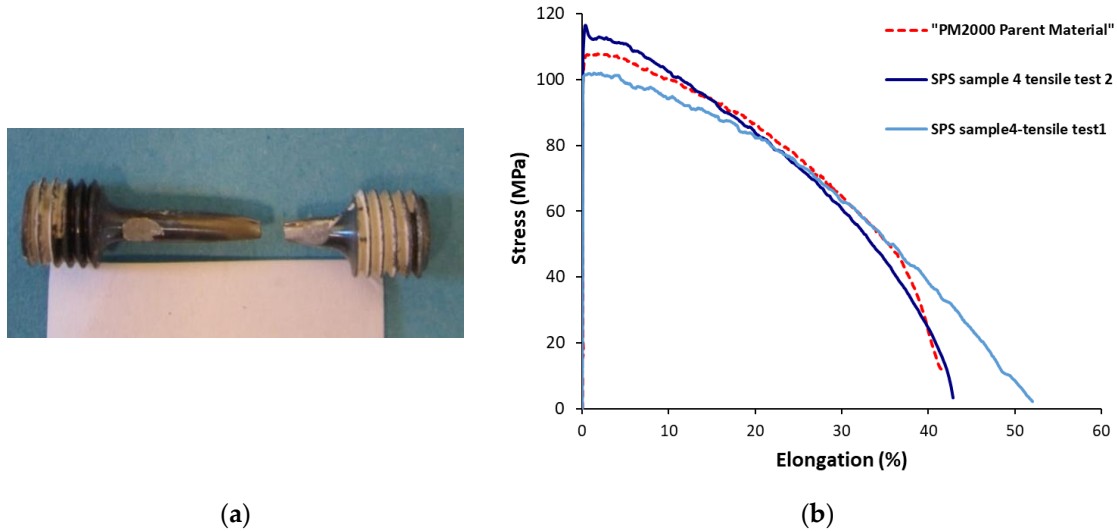

(**a**)   (**b**)

**Figure 11.** (**a**) Feature photographs of cylindrical tensile specimen (Sample 4-tensile test 2) following mechanical testing alongside (**b**) stress–strain curves at 750 °C for Sample 4 (cylindrical and flat samples) and the base PM2000 alloy in the as-received state.

Consequently, Sample 4, joined at 1100 °C with a very short dwell time, exhibited an interface that appeared clean, straight, and undamaged, even after the tensile test at 750 °C, as shown in the micrograph (Figure 12a) of the joint of the flat specimen seen in Figure 12a,b.

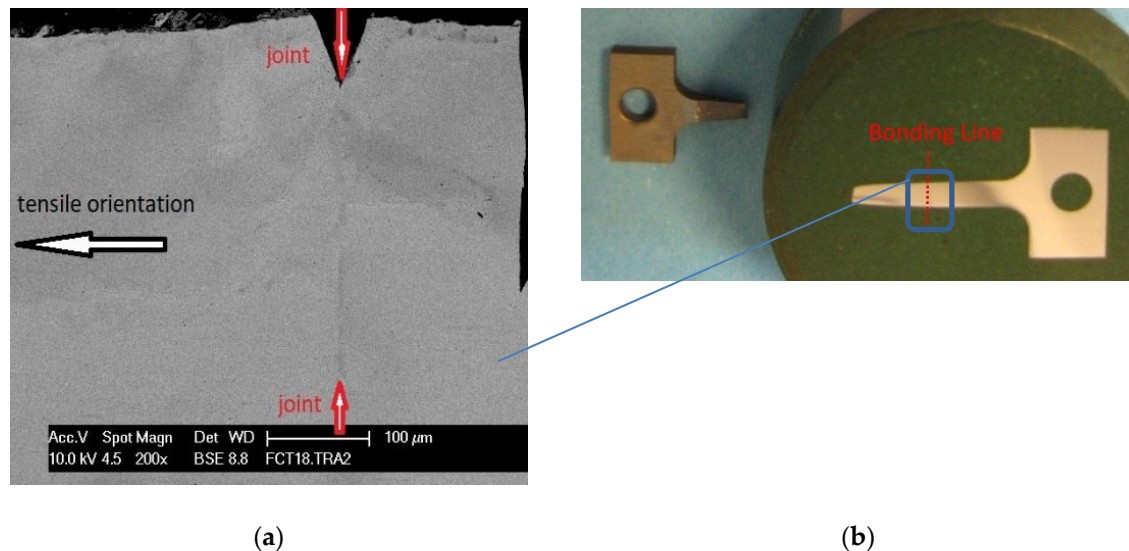

**Figure 12.** (**a**) Micrograph obtained by SEM of the bonding interface of a flat specimen from Sample 4 tested at 750 °C, (**b**) Location of the assembly area on the tensile specimen which is observed by SEM.

Nevertheless, the tensile tests conducted at 20 °C and 750 °C indicate that this joint achieved an interesting value of maximum strength. This study clearly demonstrates the potential to join ODS PM2000 steel to itself using a SPS technique.

## 4. Conclusions

Twenty wt.% chromium ODS steel PM2000 exhibiting a microstructure of coarse grain was joined to itself via SPS. Different surface preparation techniques, joining temperature, and dwell times were tested. Microstructural observations of the bonding interface were carried out and tensile tests were performed on various samples at room temperature and at 750 °C. The main results are as follows:

- Successful joining was achieved with samples that had lapped joining surfaces.
- Micrographic observations revealed some very clean joining areas. SEM and EBSD imaging of the well-bonded interface revealed recrystallized small micron-sized grains that were randomly oriented.
- The sample prepared with a lapped surface and joined by SPS at 11,000 °C during 3 min exhibited nearly the same tensile behaviour as the base material at elevated temperatures and did not fracture at the joint, but rather failure occurred in the bulk material.

This shows the importance of surface roughness and surface preparation. Indeed, the joining of ODS steel involving the raw machining of its surfaces did not result in a junction free of defects and unassembled areas. The ODS steel joined using SPS with ground surfaces did not exhibit sufficient mechanical strength. The tensile strength at 750 °C reached the nominal values (i.e., 114 MPa) of the base material, but its ductility did not exceed 8% despite the presence of many defects. However, SPS-joining of ODS steel with lapped surfaces demonstrated good performance during tensile testing at 750 °C with failure occurring in the base metal and not at the interface (i.e., an ultimate tensile strength of 101 MPa and a total elongation of 53%). The results of the mechanical tests clearly indicate that these samples joined using SPS gives rise to satisfactory mechanical strength.

**Author Contributions:** Methodology F.N., J.C.; Modelling, M.A.; Formal analysis, F.N., J.-C.N., J.C. and C.C.; Writing—original draft preparation, F.N., J.-C.N. and C.C.; Writing—review and editing, J.-M.G., M.A. and F.B.; Supervision, F.B. and J.-M.G. All authors have read and agreed to the published version of the manuscript.

**Funding:** The direction of the Nuclear Energy Department of the CEA is acknowledged for their financial support through the ASTRID project.

**Acknowledgments:** The authors would like to thank the MATEIS laboratory in INSA Lyon. We would also like to acknowledge the advice provided by G. Bonnefont (INSA Lyon) and G. Fantozzi (INSA Lyon). Specific thanks to N. Al-Mufachi from AEC for his valuable comments.

**Conflicts of Interest:** The authors declare no conflicts of interest.

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
