# Peer review of "Joining of Oxide Dispersion-Strengthened Steel Using Spark Plasma Sintering"

_metals, doi:10.3390/met10081040_

Round 1

Reviewer 1 Report

1) It is necessary to clearly set the goal of the study! Now it is too vague and therefore the conclusions are general. The third paragraph in the conclusion should be reduced - it casts doubt on the manuscript materials, since it emphasizes its shortcomings.

2) the summary and conclusions should contain more quantitative results and exclude words such as “good” and “satisfactory”.

3) in figure 1 you need to increase the scale, in figure 2 you need to insert a scale, and in figures 4 and 5 you need to delete the empty cells on the right.

4) the correct title of the article: "a preliminary study of compounds of oxide dispersion-hardened (ODS) steel by the method of spark plasma sintering."

Author Response

Dear colleague, I would first like to thank the reviewers for their in-depth review of our paper. Their various suggestions were very useful for us to improve our article. All their comments have been taken into account and have been added to the revised document attached. with my best regards F. Bernard

Reviewer 2 Report

Your manuscript is a great start to a paper on an important topic. I have some suggestions to improve the manuscript

  • In the abstract, please state spark plasma sintering before using the acronym.
  • Please include some references on the first statement of the introduction, especially for "produce less radioactive materials." How can this be true? The weak nuclear force is responsible for radioactive decay, so higher temperatures wouldn't lead to less waste right?
  • For figures 1 - 5, please remove the boxes surrounding the figures and format all the figures like figure 7 - 9. There is a big space in the two parts of figure 6, please remove.
  • Reformat table 2 to the correct format. Please work with the editor to format your table with the figure within or without.
  • Lines 130 - 132 need to be deleted. I believe these lines are from the MDPI manuscript formatting document.
  • Figure 6b, the EBSD, needs a scale and a legend for the color. Clearly, the grains near the boundary are oriented, but a color legend indicating this may elicit additional conversation about your work.
  • Line 213, please restate samples 2 and 3. You labeled them N°2 and N°3.

Thank you and good luck.

Author Response

Dear colleague, I would first like to thank the reviewers for their in-depth review of our paper. Their various suggestions were very useful for us to improve our article. All their comments have been taken into account and have been added to the revised document attached. best regards F. Bernard

Reviewer 3 Report

The subject of the paper is important in the field of joining technology (e.g. for parts in future nuclear power plants) and also of interest to the readers of Metals. The paper adds significant new insight with respect to successful joining.

The manuscript is basically well readable but further improvement is required. There are some typos and many peculiarities in the style throughout the manuscript, some ending up in errors. Try to be more precise and concise. Examples, to mention just a few:

- Line 27, comma in wrong place

- Line 32, "A … techniques"

- Line 44, "SPS technology is used to produce “in a short time” various types of materials". The authors want to express that the sintering time during SPS is shorter than for traditional methods (e.g. hot pressing) preventing unwanted grain growth. The question is not how quick a material is produced.

- Line 48, "a high current a uniaxial pressure" ???

- Lines 130-132, the authors forgot to remove the instructions for authors.

- Line 141, what is a machined orientation?

The authors jump from section 3 to section 5.

At this point it is important to emphasize that all authors, according to the rules of the journal, need to read the manuscript. They might want to help the first author (the author who wrote the draft) improving the text.

In the introduction, a more detailed review of former efforts to fabricate ODS steels from powders by means of SPS is missing. Based on this, the authors are expected to work out more clearly, what can be learned for joining and which additional challenges are encountered when SPS is used as a joining technology for two compacted samples.

The conclusions of the paper are reasonable. However, the data base of the study is not very rich. So any information that might help to strengthen the conclusions is welcome, e.g. better statistics, testing of other orientations, higher-resolution micrographs.

Author Response

Dear colleague, I would first like to thank the reviewers for their in-depth review of our paper. Their various suggestions were very useful for us to improve our article. All their comments have been taken into account and have been added to the revised document attached. Best regards F. bernard

Round 2

Reviewer 1 Report

After revision, the manuscript is significantly improved.

Author Response

Thanks for your comments.

Kind regards